# Short and Concise Peer-to-Peer Teaching—Example of a Successful Antibiotic Stewardship Intervention to Increase Iv to Po Conversion

**DOI:** 10.3390/antibiotics11030402

**Published:** 2022-03-17

**Authors:** Johannes Wild, Bettina Siegrist, Lukas Hobohm, Thomas Münzel, Thomas Schwanz, Ingo Sagoschen

**Affiliations:** 1Department of Cardiology, Cardiology I, University Medical Center Mainz, Johannes Gutenberg-University Mainz, 55131 Mainz, Germany; lukas.hobohm@unimedizin-mainz.de (L.H.); tmuenzel@uni-mainz.de (T.M.); sagosche@uni-mainz.de (I.S.); 2Center for Thrombosis and Hemostasis (CTH), University Medical Center Mainz, Johannes Gutenberg-University Mainz, 55131 Mainz, Germany; 3Department of Pharmacy, University Medical Center Mainz, Johannes Gutenberg-University Mainz, 55131 Mainz, Germany; bettina.siegrist@unimedizin-mainz.de; 4Institute of Medical Microbiology and Hygiene, University Medical Center Mainz, Johannes Gutenberg-University Mainz, 55131 Mainz, Germany; thomas.schwanz@unimedizin-mainz.de

**Keywords:** antibiotic stewardship, defined daily doses, antibiotic prescribing, antibiotic use, peer-to-peer teaching

## Abstract

Antibiotic stewardship (ABS) programs aim to combine effective treatment with minimized antibiotic-related harms. Common ABS interventions are simple and effective, but their implementation in daily practice is often difficult. The aim of our study was to investigate if a single, short, peer-to-peer teaching intervention (junior doctor to junior doctor) during clinical routine can successfully improve antibiotic prescriptions. We performed a quasi-experimental before–after study on a regular care cardiology ward at a large academic medical center in Germany. We evaluated antibiotic use metrics retrospectively and calculated defined daily doses (DDD) with the anatomical therapeutic chemical/DDD classification system of the World Health Organization. We hypothesize that the over-representative use of intravenous administration is a potentially modifiable target, which can be proven by antibiotic use metrics analysis. After a single peer-to-peer teaching session with a focus on indications for iv to po conversion, the normalized percentage of intravenous compared to oral administration significantly decreased (from 86.5 ± 50.3% to 41.4 ± 70.3%). Moreover, after the intervention, antibiotics with high oral bioavailability were no longer administered intravenously at all during the following quarter. Our results indicate that even a single peer-to-peer training session is highly effective in improving the iv to po conversion rate in the short term.

## 1. Introduction

It is common knowledge that increased antibiotic resistance directly leads to increased morbidity and mortality [1,2,3]. Furthermore, antibiotic resistance is a known global driver of healthcare costs [4,5,6]. Experience in hospitals shows that despite existing knowledge, there is still great potential to improve the antibiotic prescription practice to prevent or at least limit the emergence of multidrug-resistant pathogens [2,7]. In this regard, antibiotic stewardship (ABS) plays an increasingly important role in everyday clinical practice. ABS was once defined as “the optimal selection, dosage, and duration of antibiotic treatment that results in the best clinical outcome in the treatment or prevention of infection, with minimal toxicity to the patient and minimal impact on subsequent resistance” [8].

Designing programs and interventions to achieve this goal is an extremely multifaceted process. Therefore, selecting the most appropriate measures for each situation remains a key challenge. The World Health Organization (WHO) provides support for clinicians who are part of ABS teams worldwide by identifying the most commonly used stewardship measures and by contributing considerations for their implementation [9]. Moreover, the WHO and others [10,11] give recommendations about what a stepwise approach for a successful ABS intervention could look like. We combined this approach with the familiar quality improvement methodology of the PDCA (“plan, do, check, act”) cycle [12,13] to pilot our stewardship intervention.

Step 1: Plan—Assessment of antibiotic use. The very beginning of most ABS interventions is challenging. A detailed analysis is crucial for addressing the current state (identifying problems regarding antibiotic use) and defining realistic goals based on available resources. For care provision (quality or performance) outcome measures [14], a detailed assessment of antibiotic use prior to the design of an intervention greatly facilitates the final evaluation and represents a logical first step. To depict antibiotic use, reliably measured antibiotic use metrics such as defined daily doses (DDD), days of therapy (DOT), or the number of prescriptions per capita should be obtained and used as the backbone for further evaluations of ABS interventions [14].

Step 2: Do—Designing an ABS intervention that respects the limited resources available and meets the prevailing demands of the institution. Whereas the first step does not differ among various ABS measures, the interventions themselves are very different. As broad as the spectrum is, clinician education and training is a central and promising component of all structured ABS programs and most ABS measures [15,16]. Therefore, our overall goal was to implement a quick and concise peer-to-peer teaching intervention as an extension of the current ABS measures in our department. Although we had set the form of the envisaged ABS intervention, the exact content and target of the measure was specified by the initial assessment of antibiotic use in step 1. 

Step 3: Check—Re-Evaluation of antibiotic use metrics and the identification of possible intervention-related problems. The main part of this step is identical to the first analysis in step 1. Furthermore, additional aspects of the intervention—which are not captured by use metrics—could be detected by direct feedback mechanisms.

Step 4: Act—Implementation of the results in clinical practice. In the last step, the knowledge gained during the course of the intervention needs to be implemented in daily clinical practice, i.e., in the daily medical routine of all staff members. A successful ABS intervention can be a nice individual success, but a successful ABS program does not only consist of singular individual measures, but of a combination of different measures.

Using this approach, we performed a quasi-experimental before–after study (a study that does not use randomization to allocate the stewardship intervention but rather uses as controls different time periods and sites [14]) in the clinical setting of regular care wards of the center for cardiology at a large, urban, academic medical center in Germany. The overall question was whether a short peer-to-peer teaching intervention (junior doctor-to-junior doctor) during clinical routine could successfully improve antibiotic use. 

## 2. Results

### 2.1. Step 1—Plan

#### 2.1.1. Comparative Characterization and Identification of Determinants of Antibiotic Use

To identify possible intervention targets for a new ABS measure, we performed a detailed analysis of our department’s antibiotic consumption from 2018 to 2020. During this period, we analyzed the antibiotic use of 106,423 patient days from 26,773 cases, resulting in an average length of stay (LOS) of 3.98 days per case. In total, the regular wards of our department used 32 ± 2.7 RDDs per 100 patient days, which estimates that 32% of the inpatients received one RDD every day. To classify our values, we compared our data with published amounts of antibiotic use from other German hospitals [17]. In comparison with consumption figures from other university and non-university hospitals, the regular care wards of our department used more narrow spectrum penicillin and aminopenicillin/BLI-combinations but fewer substances from all other antibiotic groups, which resulted in no differences in the overall use of antibiotics (Figure 1A). 

In the next step, we aimed to identify the predictors of antibiotic use in our department. First, we could find a highly significant positive correlation between the DDD per 100 cases and patient days/case (Figure 1B). This finding clearly reflects the fact that patients with a longer time in hospital need more antibiotics per case than patients with a shorter time in hospital per case. Moreover, DDD/100 patient days correlated with patient days per case without reaching significance (Figure 1C). A closer evaluation revealed that DDDs/100 patient days of the most commonly prescribed oral aminopenicillin/BLI-combination, Amoxicillin + Clavulanic acid, did not correlate with patient days per case at all, whereas the iv-equivalent Ampicillin/Sulbactam showed a positive correlation (Figure 1D) without reaching significance. For the treatment of hospital-acquired infections, German guidelines recommend the use of broad-spectrum penicillins with an extended Gram-negative spectrum (e.g., Piperacillin/Tazobactam) or carbapenems (e.g., Meropenem) [18]. We were able to demonstrate a significant positive correlation between DDDs/100 PDs and PDs/case for these substances: patients who stayed longer in our clinic per case needed more second-line antibiotics (Figure 1E). Taken together, the LOS per case positively correlated with the use of broad-spectrum penicillin and carbapenems but not oral aminopenicillin in our institution. 

#### 2.1.2. Identification of Excessive Iv Administration Reflecting Low Iv to Po Conversion Rates as Target for an ABS Intervention

The evaluation of antibiotic use metrics revealed a highly significant predominance of iv antibiotic administration (Figure 2A) in our institution. Overall, 75 ± 6.73% of all DDDs on the regular care wards (outside an emergency or intensive care setting) were given intravenously (50.2 ± 11.3 DDDs iv vs. 16.1 ± 5.8 DDDs po), despite the clear data and guideline recommendations that oral administration is preferable—whenever possible—and a fast iv to po conversion is safe in defined circumstances and improves not only patient but also economic outcomes [19,20,21]. We identified this finding as a reasonable and relevant target for an ABS intervention implementing the intended peer-to-peer teaching session. To select the ward for the intervention, we compared IV rates (Figure 2B) and total intravenously administered DDDs/100 patient days of all four wards and identified significantly higher use on ward C (66.1 ± 2.6 iv DDDs on ward C vs. 48.6 ± 2.9 on ward A, 45.2 ± 4.4 on ward B, and 40.8 ± 11.0 on ward D) (Figure 2C). We next analyzed the intravenously given DDDs per 100 PDs of all five substances, which can be equivalently given via either po or iv (Figure 2D). 

We detected a significantly higher use of the aminopenicillin/BLI-combination Ampicillin/Sulbactam on ward C (24.9 ± 2.3 DDDs on ward C vs. 21.8 ± 1.5 on ward A, 15.2 ± 3.6 on ward B, and 14.3 ± 2.2 on ward D). Iv use of Clarithromycin and Levofloxacin were higher on ward A, whereas the DDDs of iv Ciprofloxacin and Clindamycin did not differ between wards. Due to the significant differences in the total iv DDDs, which were caused by the more frequent use of Ampicillin/Sulbactam, we selected ward C for our ABS intervention. 

### 2.2. Step 2: Do: Design and Implementation of a Single Quick and Simple Peer-to-Peer Teaching Session to Foster Iv to Po Conversion 

After analyzing antibiotic metrics, we found that low iv-to-po conversion rates were a major problem associated with ABS on regular care wards at our institution. We implemented peer-to-peer education on the ward with the highest number of DDD per 100 patient days. The intervention consisted of a 10 min slide-based teaching session delivered by a resident who was not a member of the regular ward staff (the slides are available in Appendix A). Because pulmonary and urinary tract infections (UTI) were the main reasons for antibiotic prescriptions, the teaching session focused on antibiotic treatment guidelines for these conditions and provided guidance on iv-to-po switching during treatment. In addition, bioavailability considerations for Clarithromycin, Levofloxacin, and Clindamycin were briefly recapped. The resident-in-training presented the slides in the ward physician’s office during the regular workday. The junior doctors of the ward were neither aware of the fact that the presentation was part of a study, nor that the metrics of antibiotic use were monitored to prevent bias. The ABS intervention was implemented before the last quarter of 2020 (Figure 3). On 1 January 2021, due to the regular rotation of residents, the entire staff of the ward changed, so that new physicians took over the duty on the ward who had not been instructed in the demonstrated peer-to-peer intervention. This allowed us to use the data from the first two quarters of 2021 as an additional control group for the effectiveness of our intervention. 

### 2.3. Step 3: Check: Antibiotic Use Metrics Prove Significant Reduction in Iv Antibiotic Administration after Peer-to-Peer Teaching

We analyzed the ABS metrics of the wards without our ABS intervention and the ward with peer-to-peer teaching at three different time points: before the intervention (2020 Q1–3), after the intervention (2020 Q4), and the two quarters after the new staff had taken over duties on the ward where the intervention had taken place (2021 Q1–2). After the intervention, normalized iv rates of all antibiotics significantly decreased (Figure 4A) on the ward where the junior doctors had been instructed (from 86.5 ± 50.3% to 41.4 ± 70.3%), whereas iv consumption rates were not altered on the wards without ABS intervention (from 119.3 ± 42.8% to 107.1 ± 48.6%). Notably, the iv rates increased again on the ward after the educated staff had left (from 41.4 ± 70.3% to 73.5 ± 61.3%). 

Levofloxacin, Ciprofloxacin, and Clindamycin were the key reasons for the observed effect (Figure 4B). The high oral bioavailability of all three substances was highlighted in the ABS teaching session. As the effect immediately disappeared after staff rotation, we interpret our findings to suggest that, in the short term, the peer-to-peer teaching was the reason why the substances were no longer administered intravenously. Combined po and iv use of Ampicillin/Sulbactam and Ciprofloxacin also decreased after ABS intervention (Figure 4C). Furthermore, not only did the iv rates change, but so did the average DDDs of all antibiotics. These were significantly higher in ward C before the intervention (3.9 ± 6.4 vs. 2.2 ± 3.5), but not in the quarters after the intervention (3.7 ± 5.8 vs. 2.9 ± 4.5 and 3.6 ± 5.4 vs. 2.9 ± 4.1) (Figure 4D). 

## 3. Discussion

Education is considered an essential component and a core element in most—some authors even say every [16]—ABS program. Educational initiatives largely target individuals with prescriptive authority [22] and are most frequently part of concomitant stewardship interventions as a form of prospective audit and feedback [16], especially multidimensional initiatives, which engage physicians in an educational process to effectively reduce unnecessary antibiotic use [23]. This clear evidence formed the basis for our approach to adding peer-to-peer teaching as another element to our already existing ABS program.

As the focus and target of all ABS projects can be summarized as improvements in the quality of treatment and patient care, it makes perfect sense to rely on established quality management (QM) procedures for guiding ABS projects. In recent years, QM methods such as the Plan–Do–Check–Act (PDCA) cycle have become increasingly popular in the healthcare sector and have proven their effectiveness [24]. The Plan–Do–Check–Act cycle is one of the most popular methods for structuring the iterative development of changes. It can further be expanded by the so-called FOCUS approach (find a process to improve, organize a team, clarify knowledge, understand causes of process variation, and select the process improvement) [25], which complements the planning phase of the PDCA cycle. Translated into the setting of ABS interventions, a careful evaluation of the ongoing antibiotic use metrics is a key factor in finding a process to improve and to pave the path for a new ABS measure. Moreover, evaluation of the antibiotic use metrics is also crucial in measuring the quality and multidimensional impacts of the measure [26]. 

The initial metrics analysis in our department revealed the fact that the overall use of antibiotics on our regular care wards did not significantly differ from their average use in other German hospitals. At first glance, this shows that there are no serious problems with increased antibiotic use at our institution; however, the comparative data were not collected on cardiology wards only, but included entire clinics such as emergency medicine, hematology, oncology, and intensive care, all of which are known to be heavy users of antibiotics [27]. We further aimed to disentangle the predictors of antibiotic use in our institution. We could give a clear positive correlation between the length of hospital stay per case and the use of broad-spectrum penicillin and carbapenems per patient day, whereas oral aminopenicillin use did not correlate at all. We interpret this—at first glance, a non-surprising finding—as quality criteria for our data. This finding reflects the assumption that patients who stay longer in the hospital need more antibiotics per day than patients with a shorter intra-hospital time. This finding might not be surprising as patients with a longer hospital stay per case provide a higher disease severity than patients with a shorter intra-hospital time. In the cardiologic setting of our study, patients hospitalized for prescheduled elective procedures such as catheter ablation for atrial fibrillation or planned percutaneous coronary interventions are hospitalized for significantly shorter periods than patients administered via the emergency department due to pneumonia, decompensated heart failure, or cardiac arrest and rarely need antibiotic therapy at all. In addition, the hospital LOS for community acquired infections (e.g., pneumonia), is known to be significantly shorter than for hospital-acquired infections [28]. The clear difference in correlation between second-line antibiotics and oral aminopenicillin/BLI combinations underlines this assumption. Given the fact that oral and iv aminopenicillins + BLI are widely used in Germany to treat community-acquired respiratory infections (also supported by the German guidelines [29]), this might indicate an increased LOS for those patients in the hospital. In contrast to the aminopenicillin/BLI-combinations targeting community-acquired infections, the intention to treat hospital-acquired infections might be the key driver for the significant correlation of broad-spectrum penicillin and carbapenems with LOS. 

As interesting as this fact is, it did not constitute a modifiable cause for our planned ABS project. However, in further analyses of the antibiotic use metrics of our institution, we detected a significant iv application rate as proof of low iv-to-po conversion rates in the regular care wards of our institution. 

The benefits of early conversion from intravenous to oral medication are well studied [30,31]. Several observational studies clearly demonstrated that early conversion to oral medication did not result in higher rates of treatment failure [32] but in shorter lengths of stay, e.g., in patients with pneumonia [33,34,35]. In addition to better patient outcomes by a timely conversion of iv medications to highly bioavailable oral forms that can minimize IV line-related infections and adverse effects of administration, fast iv-to-po conversion also provides tremendous economic benefits [20,36,37]. 

The strength of our study consists of the very simple, extremely short, and therefore cost-effective design of our intervention. Whereas most published educational interventions consist of different educational interventions such as educational sessions, audits, and feedback, our peer-to-peer teaching was only based on one single presentation. Furthermore, the educated junior doctors were not aware of the fact that they were part of an interventional study or that the antibiotic use metrics of their ward were surveilled to avoid any bias. All the more pleasing is the significant decrease observed in the number of intravenously administered antibiotics on the ward after the intervention (Figure 4A), which was also accompanied by a significant decrease in all DDDs prescribed (Figure 4D). 

Our study has limitations that we want to address. We can only present the data of a single-center study with only one selected ward as an intervention group. As depicted in Figure 1B, the patient days per case differ between the different wards of our institution, so the effect of the intervention could differ between every ward. Furthermore, we can only provide short-term data of one quarter and therefore cannot estimate if the observed effect diminishes over time. To evaluate whether a single educational intervention can really achieve sustainable effects, a longer follow-up with unchanged staffing would be necessary. The short observation period also causes the limitation that we can only speculate about seasonal effects. We had planned the intervention before the onset of the COVID-19 pandemic and had selected the last quarter of 2020 (the months October, November, and December) as the start for the intervention, as we expected a high usage of antibiotics driven by seasonal effects. Due to pandemic-related measures, the rate of non-COVID-19 respiratory diseases was significantly reduced in 2020 [38,39], so the selected period might not be representative. As we performed the study on a cardiology regular care ward where no COVID-19 patients were treated, patients with COVID-19 are not part of our analysis at all. Nevertheless, the general effects of COVID-19 on hospital admissions might have influenced our study [40]. 

## 4. Conclusions

Our study results indicate that even a very short and concise educational session in a peer-to-peer setting can significantly reduce iv antibiotic use. Our data contribute further evidence that education is a core element of every ABS program. The straightforward design of our intervention using established tools of quality management underlines that ABS measures do not need to be expensive or complicated to make a difference. In the hectic and stressful clinical routine, education and training often fall short. All facts taught in our ABS training are part of early training in medical school and are not newly discovered scientific facts, so we suspect a lack of ABS guidance, especially after graduation. It remains a core task of every ABS team to constantly spread and reactivate basic ABS knowledge.

## 5. Materials and Methods

### 5.1. Study Design and Data Source

We retrospectively analyzed data of antibiotic use of all four normal care cardSiology wards (wards A–D) of the Center for Cardiology at the University Medical Center Mainz (Mainz, Germany). For this purpose, we used the digital ordering system (SAP) of the University Medical Center, which tracks every drug order from the wards. Data about antibiotic use were acquired yearly for the years 2018/2019 and quarterly for the years 2020 and 2021. We obtained data about patient days and cases for the same intervals (numbers of patient days and case numbers from the different wards were provided by the controlling of the Center for Cardiology (University Medical Center Mainz) retrospectively). 

### 5.2. Measures of Antibiotic Use

As an independent comparator for quantitative and qualitative analyses of antibiotic use, we calculated defined daily doses with the ATC/DDD (anatomical therapeutic chemical/defined daily dose) classification system of the WHO (ATC/DDD Index 2022; https://www.whocc.no/atc_ddd_index/; access date 17 March 2022). Antibiotic consumption density was calculated as follows: number of daily doses (DDD) of the active ingredient/100 patient days (or 100 cases), as recommended by the WHO.

### 5.3. Design and Execution of the Peer-to-Peer Teaching ABS Intervention

By analyzing the antibiotic metrics, we identified low iv-to-po conversion rates as one of the major ABS-related issues of the regular care wards of our institution. We implemented the peer-to-peer teaching on the ward with the highest amount of DDD per 100 patient days (ward C). The intervention consisted of a 10 min PowerPoint slide-based teaching session. A junior doctor who was not part of the regular staff on the ward presented the slides in the doctor’s room on ward during the regular working day. The junior doctors of the ward were neither aware of the fact that the presentation was part of a study, nor that the metrics of antibiotic use are monitored. The slides of the PowerPoint presentation can be found in Appendix A; the time schedule is presented in Figure 4.

### 5.4. Statistical Analysis

We performed statistical analysis with GraphPad Prism software (version 9; GraphPad Software, Inc.). In the case of a normal distribution (analyzed by Kolmogorow–Smirnow test or Shapiro–Wilk test), a two-way ANOVA test with Tukey’s post hoc test or Student’s *t*-test were applied. If there was no normal distribution, a Kruskal–Wallis test with Dunn’s multiple comparison or a comparison of selected columns was used. Furthermore, simple linear regression was calculated as indicated. *p* < 0.05 was required to achieve statistical significance. Mean values are shown with SEMs. 

## Figures and Tables

**Figure 1 antibiotics-11-00402-f001:**
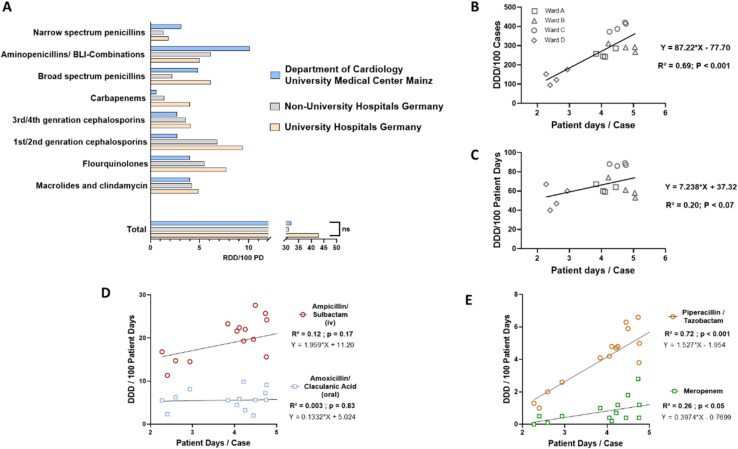
Characterization of antibiotic use and correlation of patient days per case with antibiotic use. (**A**) Comparison of antibiotic use of all regular wards of our institution (Department of Cardiology University Medical Center Mainz) as mean recommended daily doses (RDDs) per 100 patient days (PD) (years 2018–2021) with other university/non-university hospitals (data for comparison from [17]); *n* = 8 per group; two-way ANOVA with Tukey’s multiple comparison test; ns = not significant. (**B**) Simple linear regression analysis of DDD of all antibiotics/100 cases and patient days per case of all four regular care wards of our institution (yearly; 2018–2020). (**C**) Simple linear regression analysis of DDD of all antibiotics/100 patient days and patient days per case (yearly; 2018–2020). (**D**) Simple linear regression of DDD of Ampicillin/Sulbactam and Amoxicillin/Clavulanic Acid per 100 patient days and patient days per case (yearly; 2018–2020). (**E**) Simple linear regression analysis of DDD of Piperacillin/Tazobactam and Meropenem per 100 patient days and patient days per case (yearly; 2018–2020).

**Figure 2 antibiotics-11-00402-f002:**
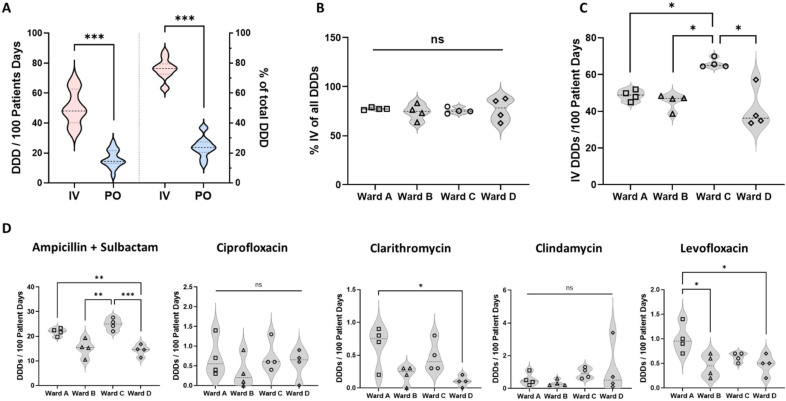
Rates of intravenous administration and discrimination between different wards. (**A**) Iv and po administered DDDS/100 PDs; *n* = 16 per group; unpaired *t*-test; *** *p* < 0.001. (**B**) Percentage of iv administration of all DDDs on all regular care wards. Two-way ANOVA with Tukey’s multiple comparison test; ns = not significant. (**C**) Total iv DDDs/100 PD. *n* = 4 per group; two-way ANOVA with Tukey’s multiple comparison test; * *p* < 0.05. (**D**) Intravenous administration of Ampicillin + Sulbactam, Ciprofloxacin, Clarithromycin, Clindamycin, and Levofloxacin. *n* = 4 per group; two-way ANOVA with Tukey’s multiple comparison test; *** *p* < 0.001, ** *p* < 0.01, * *p* < 0.05.

**Figure 3 antibiotics-11-00402-f003:**
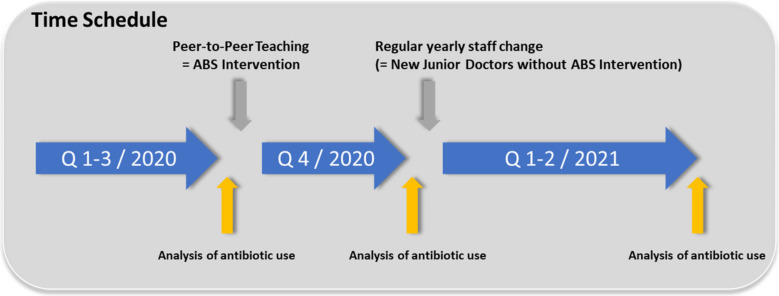
Time schedule of the ABS intervention.

**Figure 4 antibiotics-11-00402-f004:**
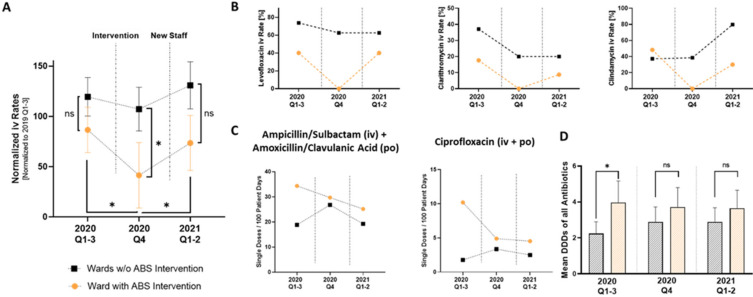
Lower iv percentages of total DDDs were driven by a total reduction in iv Levofloxacin, iv Clindamycin, and iv Clarithromycin use. (**A**) Normalized iv rates of all DDD of Levofloxacin, Ciprofloxacin, Clindamycin, Ampicillin/Clavulanic acid, and Ciprofloxacin on the ward, with and without ABS intervention (normalized to mean iv DDDs of Q1–3 2020 on all wards); *n* = 5 per group; paired *t*-test; * *p* < 0.05. (**B**) Mean iv rates of Levofloxacin, Clarithromycin, and Clindamycin (*n* = 1 vs. 3; no statistics). (**C**) Combined iv and po single doses/100 patient days of Ampicillin/Sulbactam and Amoxicillin/Clavulanic acid and Ciprofloxacin (*n* = 1 vs. 3; no statistics). (**D**) Time course of mean single doses of all antibiotics (*n* = 29 per group; Wilcoxon matched pairs signed rank test; * *p* < 0.05).

## Data Availability

All data are presented in the current manuscript and its Appendix A.

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
