# Peer review of "Short and Concise Peer-to-Peer Teaching—Example of a Successful Antibiotic Stewardship Intervention to Increase Iv to Po Conversion"

_antibiotics, 2022, doi:10.3390/antibiotics11030402_

Round 1
Reviewer 1 Report
I am delighted to review this manuscript, covering an important subject i.e introduction of peer-to-peer teaching as another element to antibiotic stewardship, and controlling the use of iv for antibiotics after implication of another antibiotic stewardship. Although the study is limited to one ward, and due to Covid-19 impacting the study, still the outcome is very impactful. The manuscript follows the scope of the journal Antibiotics. A detailed study has been done with a good presentation of research work.
I would recommend the article could be published in Antibiotics with minor corrections. And the authors need to address the below-mentioned queries.
(a) Introduction should be concise.
(b) The author needs to recheck all the symbols used in table 1.
(c) The author could include the English version of the presentation in the supplemental file for the common reader.
(d) The author could include a conclusion section.
(e) The author could include the following relevant reference,
(a) Wilf-Miron R, Ron N, Ishai S, Chory H, Abboud L, Peled R. Reducing the volume of antibiotic prescriptions: a peer group intervention among physicians serving a community with special ethnic characteristics. J Manag Care Pharm. 2012 May;18(4):324-8. doi: 10.18553/jmcp.2012.18.4.324. PMID: 22548692.
Author Response
Comments and Suggestions for Authors
I am delighted to review this manuscript, covering an important subject i.e introduction of peer-to-peer teaching as another element to antibiotic stewardship, and controlling the use of iv for antibiotics after implication of another antibiotic stewardship. Although the study is limited to one ward, and due to Covid-19 impacting the study, still the outcome is very impactful. The manuscript follows the scope of the journal Antibiotics. A detailed study has been done with a good presentation of research work.
I would recommend the article could be published in Antibiotics with minor corrections. And the authors need to address the below-mentioned queries.
Response: We thank the reviewer for the thorough evaluation of our manuscript and the positive feedback. We implemented all mentioned suggestions.
(a) Introduction should be concise.
Response: We thank the reviewers for the suggestion and shortened the introduction.
(b) The author needs to recheck all the symbols used in table 1.
Response: We recheck the symbols in figure 1.
(c) The author could include the English version of the presentation in the supplemental file for the common reader.
Response: We apologize for our negligence and provide a translated version of the supplementary materials.
(d) The author could include a conclusion section.
Response: We thank the reviewer for this comment. We restructured our discussion section and now provide a conclusion at the end.
(e) The author could include the following relevant reference,
(a) Wilf-Miron R, Ron N, Ishai S, Chory H, Abboud L, Peled R. Reducing the volume of antibiotic prescriptions: a peer group intervention among physicians serving a community with special ethnic characteristics. J Manag Care Pharm. 2012 May;18(4):324-8. doi: 10.18553/jmcp.2012.18.4.324. PMID: 22548692.
Response: We implemented the reference as suggested.

Reviewer 2 Report
Overall: The present study is a behavioral intervention study evaluating the impact of a peer-to-peer teaching antimicrobial stewardship intervention on antibiotic use within a German hospital. Overall, the study found the intervention was associated with significant decrease in intravenous antibiotic use. These data are informative and interesting. I applaud the efforts of the authors in their work. However, there are a couple of considerations that may be evaluated before publishing.
General: if publishing in english journal, would consider providing english translated supplementary materials. My consider changing abs to asp acronym as more common in english literature
Abstract: would consider quantifying % change observed in abstract
Page 7 Lines 273-276: Would consider adding a line related to the correlated topic of de-escalation of broad spectrum being associated with decrease length of stay in the literature
Dan Ilges, David J Ritchie, Tamara Krekel, Elizabeth A Neuner, Nicholas Hampton, Marin H Kollef, Scott Micek, Assessment of Antibiotic De-escalation by Spectrum Score in Patients With Nosocomial Pneumonia: A Single-Center, Retrospective Cohort Study, Open Forum Infectious Diseases, Volume 8, Issue 11, November 2021, ofab508, https://doi.org/10.1093/ofid/ofab508
Bohan JG, Remington R, Jones M, Samore M, Madaras-Kelly K. Outcomes Associated With Antimicrobial De-escalation of Treatment for Pneumonia Within the Veterans Healthcare Administration. Open Forum Infect Dis. 2016;4(1):ofw244. Published 2016 Dec 10. doi:10.1093/ofid/ofw244
Author Response
Comments and Suggestions for Authors
Overall: The present study is a behavioral intervention study evaluating the impact of a peer-to-peer teaching antimicrobial stewardship intervention on antibiotic use within a German hospital. Overall, the study found the intervention was associated with significant decrease in intravenous antibiotic use. These data are informative and interesting. I applaud the efforts of the authors in their work. However, there are a couple of considerations that may be evaluated before publishing.
Response: We thank the reviewer for the careful evaluation of our manuscript and the positive feedback. We could implement all suggestions.
General: if publishing in english journal, would consider providing english translated supplementary materials. My consider changing abs to asp acronym as more common in english literature
Response: We thank the reviewer for his suggestion and apologize for our negligence. Now, we provide a translated version of the supplementary materials.
Abstract: would consider quantifying % change observed in abstract
Response: We added the quantification in the abstract.
Page 7 Lines 273-276: Would consider adding a line related to the correlated topic of de-escalation of broad spectrum being associated with decrease length of stay in the literature
Dan Ilges, David J Ritchie, Tamara Krekel, Elizabeth A Neuner, Nicholas Hampton, Marin H Kollef, Scott Micek, Assessment of Antibiotic De-escalation by Spectrum Score in Patients With Nosocomial Pneumonia: A Single-Center, Retrospective Cohort Study, Open Forum Infectious Diseases, Volume 8, Issue 11, November 2021, ofab508, https://doi.org/10.1093/ofid/ofab508
Bohan JG, Remington R, Jones M, Samore M, Madaras-Kelly K. Outcomes Associated With Antimicrobial De-escalation of Treatment for Pneumonia Within the Veterans Healthcare Administration. Open Forum Infect Dis. 2016;4(1):ofw244. Published 2016 Dec 10. doi:10.1093/ofid/ofw244
Response: We thank the reviewer for the suggestion and implemented the references in our manuscript.
.

Round 2
Reviewer 1 Report
I would like to thank the authors of the manuscript for the consideration of the reviews’ comments and for making the necessary corrections/changes accordingly. The author has made necessary corrections and modifications to the manuscript as suggested by the reviewers which remarkably enhanced the overall quality of the manuscript with a clear presentation of results and proper discussion.
Some recommendations of the journal for manuscript presentation are ignored but I believe that the editor will point out these problems.
I would recommend the article could be published in Antibiotics in the present form, and the authors need to correct the following minor errors.
- In supplementary: Page 2; space needed “300mg” to “300 mg” etc.
- In supplementary: Page 3 and subsequent pages; Start the sentence with a capital letter.
- The author needs to enhance the resolution of Figures 1, 2, 3, and 4.